# Sometimes I am a Tree: Data Drives Unstable Hierarchical Generalization

## Abstract

Neural networks often favor shortcut heuristics based on surface-level patterns. Language models (LMs), for example, behave like n-gram models early in training. However, to correctly apply grammatical rules, LMs must instead rely on hierarchical syntactic representations rather than on surface-level heuristics derived from n-grams. In this work, we use cases studies of English grammar to explore how latent structures in training data drives models toward improved out-of-distribution (OOD) generalization. We then investigate how data composition can lead to inconsistent behavior across random seeds. Our results show that models stabilize in their OOD behavior only when they commit to either a surface-level linear rule or a hierarchical rule. The hierarchical rule, furthermore, is induced by grammatically complex sequences with deep embedding structures, whereas the linear rule is induced by simpler sequences. When the data contains a mix of simple and complex examples, potential rules compete; each independent training run either stabilizes by committing to a single rule or remains unstable in its OOD behavior. We also identify an exception to the relationship between stability and generalization: Models which memorize patterns from homogeneous training data can overfit stably, with different rules for memorized and unmemorized patterns. While existing works have attributed similar generalization behavior to training objective and model architecture, our findings emphasize the critical role of training data in shaping generalization patterns and how competition between data subsets contributes to inconsistent generalization outcomes.

## 1 Introduction

Neural networks often learn shortcut heuristics which reflect simple, surface-level patterns in data. In the case of language models (LMs) trained on next-token prediction objectives, this simplicity bias can lead models to behave like n-gram models, relying heavily on local dependencies without fully capturing the deeper, more complex structures of language (Choshen et al., 2022; Geirhos et al., 2020; Saphra & Lopez, 2018). However, LMs are also capable of breakthroughs in generalization, shifting from these simple heuristics to more sophisticated behaviors (Choshen et al., 2022; Chen et al., 2023; McCoy et al., 2020). Such transitions suggest that under certain training conditions, LMs can eventually overcome spurious shortcuts and use linguistic structures to generalize beyond surface-level patterns. Previous works often attribute this ability to model architecture and training objectives (Ahuja et al., 2024; McCoy et al., 2020). In this work, we investigate how *data* characteristics influence the generalization rules learned, especially when multiple solutions fit the training data equally well. We also examine the instabilities associated with generalization behaviors.

To understand when and why a model favors learning latent structures over surface-level heuristics, we use case studies in learning English grammar rules. Grammatically correct sentences in English must follow a set of rules that operate on a sequence's latent tree-like structure (Chomsky, 2015; Crain & Nakayama, 1987). When trained on next-token prediction, an LM may approximate these rules from surface-level statistics, acting as an n-gram model by applying a *linear rule*. However, such a model struggles to generalize to unseen grammatical patterns. Figure 1 (*bottom right*) shows that a LM can use a linear bigram model to capture the relationship between a subject noun and its verb by *inflecting* the verb with the same plurality as the subject. This LM would fail to generalize when a *distractor* noun, e.g., from a prepositional phrase, appears between subject and verb. In contrast,

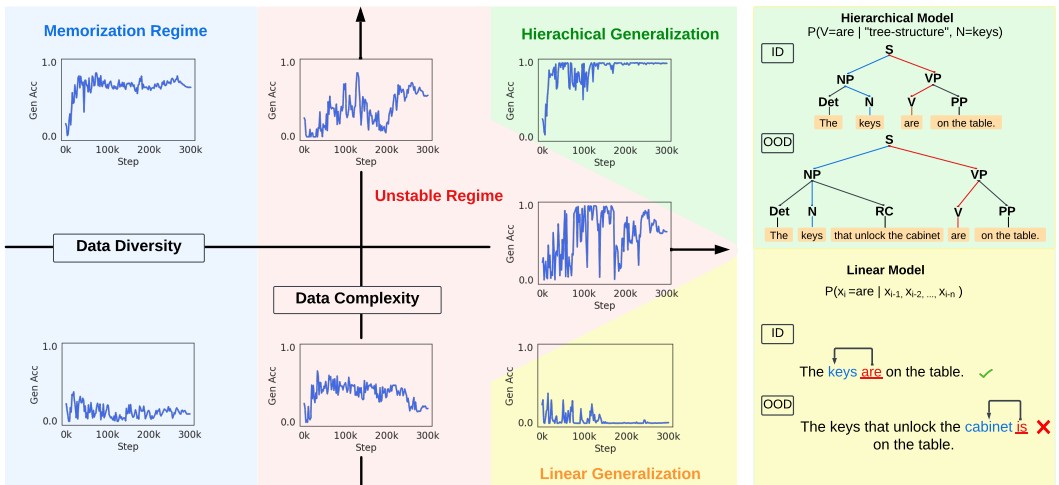

Figure 1: **Data plays a critical role in generalization behaviors and training stability.** *Left:* Along the data diversity x-axis, Low data diversity (as measured by variation in syntactic structure) leads the model to memorize unreliable sample-specific patterns, whereas high data diversity promotes commitment to a general rule. Along the data complexity y-axis, high data complexity (i.e., with more center embeddings) induces the hierarchical rule, while simpler data (right-branching sentences) induces the surface-level linear rule. Mixing these data types results in unstable OOD training behaviors. *Upper Right:* A model that captures hierarchical structure can generalize grammatical rules OOD by correctly identifying the subject as the noun closest to the root on the syntax tree graph. *Lower Right:* A model that uses the linear rule will treat the most recent noun as the target verb's subject and thereby fail to generalize to unseen sentence compositions.

Figure 1 (*upper right*) shows a model that instead uses a latent tree structure can learn the correct syntactic rule (i.e., the *hierarchical rule*), enabling it to generalize to novel sentence compositions.

Building on previous work (McCoy et al., 2018; 2020; Ahuja et al., 2024), we use two tasks—question formation and tense inflection—to investigate whether the model learns the hierarchical rule or defaults to the surface-level linear rule. We train models on ambiguous data, which is compatible with both rules, and evaluate them on out-of-distribution (OOD) data which is compatible only with the hierarchical rule. We first find that a preference for OOD hierarchical generalization is induced by training on samples with center embeddings, where the subject is modified by an relative clause. This result mirrors a celebrated claim from linguistics (Wexler, 1980) that center embeddings are responsible for human syntax acquisition.

Models trained on the same data exhibit inconsistent OOD behaviors across random seeds. By examining training dynamics, we identify a connection between training stability and rule commitment: Only runs that commit to a rule can exhibit stable OOD performance during training. Connecting back to data, we show that training dynamics can be categorized into three distinct regimes that depend on data complexity and diversity, illustrated in Figure 1 *left*. Data diversity determines whether a model learns a general rule, while data complexity determines which rule is preferred. Models trained on a mix of hierarchical-inducing samples and linear-inducing samples are most unstable during training and exhibit the largest inconsistency across random seeds.

Taken together, our findings demonstrate that data composition plays a critical role in shaping model's OOD generalization behaviors. Our contributions are as follows:

- We show that sentences with complex grammatical structure—specifically center embeddings—drive LMs to favor hierarchical syntactic representations over surface-level n-gram heuristics, enabling correct OOD generalization of grammatical rules (see Section 4).
- We demonstrate that models stabilize in OOD performance only when they commit to either a surface-level heuristic or a hierarchical rule (see Section 5).

- We show that when the training data mixes complex and simple grammatical structures, the resulting rules are inconsistent across random seeds and many models fail to stabilize OOD behavior by the end of training (see Section 5).
- We identify an exception to the relationship between stability and rule learning: Models trained on insufficiently diverse data stabilize in a memorization regime without learning either rule, highlighting another way that data can drive generalization failures (see Section 6).

## 2 RELATED WORK

We include the most relevant work in this section. For an extended discussion of related work, please refer to Appendix A.

### 2.1 SYNTAX AND HIERARCHICAL GENERALIZATION

McCoy et al. (2018) first used the question formation task to study hierarchical generalization in neural networks, showing that RNNs trained with a seq-to-seq objective exhibit limited hierarchical generalization. However, adding attention mechanisms improved performance on the generalization set. Later, McCoy et al. (2020) found that tree-structured architectures consistently induce hierarchical generalization. Petty & Frank (2021) and Mueller et al. (2022) further investigated inductive biases and concluded that, like RNNs, transformers tend to generalize linearly. This view was challenged by Murty et al. (2023), who attributed the failure of prior attempts to insufficient training, demonstrating that decoder-only transformers can generalize hierarchically, but only after in-distribution performance has plateaued. Expanding on this, Ahuja et al. (2024) showed that hierarchical generalization is achieved only with models trained on a language modeling objective. Previous work primarily attributed the source of inductive bias toward hierarchical rule to model architecture, whereas our study highlights the impact of data, and we further provide a precise measure of data complexity and data diversity. In addition, the inconsistency in generalization behaviors are observed in (McCoy et al., 2018; McCoy et al.). While they have pointed out that models trained on different random seeds can manifest very different generalization behaviors, they did not further studies the distributions of model behaviors.

Similar to our work, Papadimitriou & Jurafsky (2023) and Papadimitriou & Jurafsky (2020) also studied how training data could introduce an inductive bias to affect language acquisition. specifically identified that by pretraining models on data with a recursive structure, finetuning them on natural language yields superior performances. This finding is closely related to our conclusions around center embeddings since the center embedding structure in language is recursive in nature.

### 2.2 TRAINING DYNAMICS AND GROKKING

Grokking refers to the phenomenon where a neural network, after achieving seemingly poor performance for a long period, suddenly generalize on unseen data. Power et al. (2022) first observed this behavior in simple arithmetic tasks. Since then, the exact mechanism of grokking has been widely studied. Different from existing grokking work, we studied a different types of grokking: "structural grokking" (Murty et al., 2023). In classic grokking, the model transitions from memorization to generalization, allowing it to achieve non-trivial performance on unseen data. In structural grokking, a model transitions from the simple linear rule to the hierarchical rule, leading to non-trivial performance on OOD data. However, the findings of this study is potentially related to those in classic grokking. Zhu et al. (2024) studies the role of data and finds that grokking only occurs when training data is sufficiently large. Berlot-Attwell et al. (2023) broadly studies how data diversity and complexity leads to different generalization behaviors. Liu et al. (2022) shows that grokking can be induced with different weight norms, associating generalization with a specific goldilocks zone weight norm value. Huang et al. (2024) and Varma et al. (2023) have shown that during training, different circuits compete, and models trained on different random seeds can lead to distinct generalization behaviors depending on which circuits dominate. While these works primarily attribute generalization differences to circuit formation, our findings highlight that this competition between circuits can also destabilize learning dynamics. Importantly, we characterize the unstable regime in both data diversity and data complexity and we address connections between training stability and consistency under random variation.

Table 1: **Examples from two grammar case studies.** *Top*: In the question formation task, the model moves the main auxiliary verb to the front to form a question. *Bottom*: In the tense inflection task, the model inflects the main verb from past to present tense, while respecting subject-verb agreement.

| Dataset | Task Type | Examples |
|---------|-----------|----------|
| Question Formation | Quest (Ambiguous) | **Input:** My unicorn does move the dogs that do wait. 
 **Output:** Does my unicorn move the dogs that do wait? |
| | Quest (Unambiguous) | **Input:** My unicorn who doesn't sing does move. 
 **Linear Output:** Doesn't my unicorn who sing does move? 
 **Hierarchical Output:** Does my unicorn who doesn't sing move? |
| Tense Inflection | Present (Ambiguous) | **Input:** My zebra behind the peacock smiled. 
 **Output:** My zebra behind the peacock smiles. |
| | Present (Unambiguous) | **Input:** My zebra behind the peacocks smiled. 
 **Linear output:** My zebra behind the peacocks smile. 
 **Hierarchical output:** My zebra behind the peacocks smiles. |

## 2.3 RANDOM VARIATION

Although choices like hyperparameter settings, architecture, and optimizer all shape model outcomes, training remains inherently stochastic. Models are sensitive to random initialization and the order of training examples. Several studies (Zhou et al., 2020; D'Amour et al., 2022; Naik et al., 2018) have reported significant performance differences across model checkpoints and Zhou et al. (2020) noted that instability extends throughout the training curve. Dodge et al. (2020) found that both weight initialization and data order contribute equally to out-of-sample performance variation. Unlike prior work, which focuses on the experimental implications of random variations, we investigate the source of these training inconsistencies and link them to characteristics of the training data.

## 3 EXPERIMENTAL SETUP

The question formation task and the tense inflection task are first proposed by Frank & Mathis (2007) and Linzen et al. (2016) as canonical tasks to assess a model's language modeling ability. In this study, we use the synthetic dataset constructed by McCoy et al. (2018) (for question formation) and McCoy et al. (2020) (for tense inflection).

### 3.1 QUESTION FORMATION TASK

In the **question formation (QF)** task, a declarative sentence is transformed into a question (see Table 1) by moving the main auxiliary verb (such as "*does*" in "*does move*") to the front. Our training data permits two strategies for choosing which verb to move: (1) *move first*: a linear rule that moves the first auxiliary, or (2) *move main*: a hierarchical rule—the correct rule in English grammar—based on the sentence's syntactic structure. This syntactic structure links each word into a tree-like structure in which edges specify syntactic dependencies (e.g., subject, preposition, object), as shown in Figure 2. The model leverages this tree representation to determine which auxiliary to move.

In Table 1, the first example is considered **ambiguous** because both the hierarchical and linear rules produce the correct outcome. In contrast, the second example is **unambiguous** because only the hierarchical rule produces the correct outcome. The training data contains only ambiguous samples, while the OOD generalization set includes only unambiguous samples. If a model uses a hierarchical representation of syntax, it should achieve $100\%$ accuracy on both the in-distribution (ambiguous questions) and OOD generalization (unambiguous questions) sets. Conversely, if a model rise on linear rules, it will score $0\%$ on the OOD generalization set, but still score $100\%$ accuracy on the in-distribution set. We therefore use the model's accuracy on the OOD generalization set as a metric for hierarchical generalization.

### 3.2 TENSE INFLECTION TASK

In the **tense inflection (TI)** task, we provide the model with a sentence in the past tense, and the model transforms it into the present tense. Since past-tense verbs in English do not differentiate between singular and plural forms, the model must identify the subject to determine whether the present-tense verb should be inflected as singular or plural. The TI task tests whether the model follows the hierarchical or linear rule for subject-verb agreement. The linear rule inflects the verb

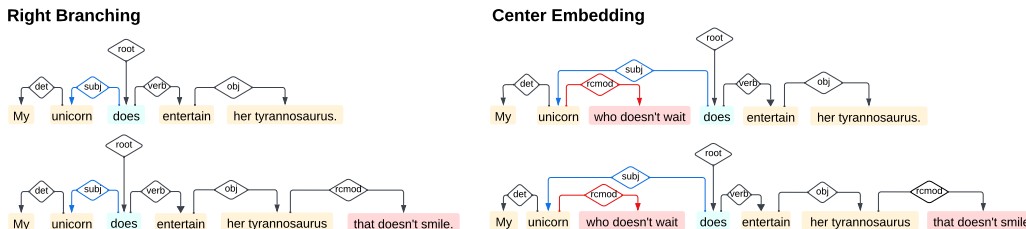

Figure 2: **Sentence Examples.** *Left:* Right-branching sentence examples. The linear progression of the main phrase is not interrupted by the relative clause. *Right:* Center-embedded sentence examples. When the relative clause modifies the subject, it interrupts the linear progression of the main clause.

based on the most recent noun, while the hierarchical rule correctly inflects the verb according to the subject. Like in the QF task, the training data contains ambiguous samples (example in Table 1), where the subject noun (i.e., "*zebra*") and the most recent noun (i.e., "*peacock*") always share the same plurality and therefore either rule produces the correct answer. The OOD generalization set includes unambiguous examples, where the subject and the most recent noun differ in plurality and therefore only the hierarchical rule produces the correct answer. Similar to the QF task, we use the model's main-verb prediction accuracy on the OOD set as a metric for hierarchical generalization.

### 3.3 Model, Data and Training

We use a decoder-only Transformer architecture with 12M parameters: 6 layers of 8 heads with a 512-dimensional embedding for QF. For TI, we use the same transformer architecture but with 4 layers. All models are trained from scratch on a causal language modeling objective for 300K steps. We use the Adam optimizer (Kingma & Ba, 2014), a learning rate of 1e-4, and a linear decay schedule. All the hyperparamter settings are directly adopted from existing works (Ahuja et al., 2024; Murty et al., 2023). We run all experiments on the same 50 random seeds. We use the original training, validation and OOD generalization data proposed by McCoy et al. (2018) and McCoy et al. (2020). To create variations on the training data, we mimic the data generation process used for the original QF and TI task. Specifically, the original TI and QF data are generated with Context-Free Grammar (CFG) using a simplified set of grammatical rules, and we reuse the same CFG rules to create variations of the training data. We use a word-level tokenizer with a vocabulary of size 72.

## 4 Data Complexity Determines Rule Preference

We begin by analyzing center-embedded sentences in Section 4.1. We then show that center-embedded sentences drives hierarchical generalization in both QF task (Section 4.2) and TI task (Section 4.3).

### 4.1 Center Embedding

Center embedding occurs when a clause—often acting as a modifier—is placed within another clause or phrase. Figure 2 (*left*) illustrates two examples of center-embedded sentences, where the embedded clause disrupts syntactic dependencies, such as the subject-verb-object relationship. Moreover, center embeddings exhibit a recursive structure: inside the relative clause, one can find the same structure as the entire sentence. In contrast, sentences without center embeddings are exclusively right branching. Right-branching structures may also include modifying clauses, but these can only be appended at the end of the main clause, maintaining its linear flow (see Figure 2, *right*). Center embedding has been central to linguistic studies on the types of data required to learn grammatical rules. According to Chomsky's generative grammar framework (Chomsky, 2015), center-embedded clauses give rise to hierarchical, tree-like syntactic structures. Additionally, Wexler (1980) posits that all English syntactic rules can be learned from "degree 2" sentences, which contain exactly one embedded clause.

While center embeddings are crucial for human language acquisition, in this study we investigate whether the same type of data can lead a LM to acquire the hierarchical grammar rule. To correctly predict the distribution of next tokens, LMs must track dependencies between sentence components. In right-branching sentences, LMs can rely on linear proximity to identify dependencies; for example, as shown in Figure 2, a simple bigram model suffices to capture the subject-verb relationship. In contrast, center embeddings introduce relative clauses of various lengths, making linear n-gram models inefficient for capturing subject-verb dependencies. Furthermore, the recursive nature means

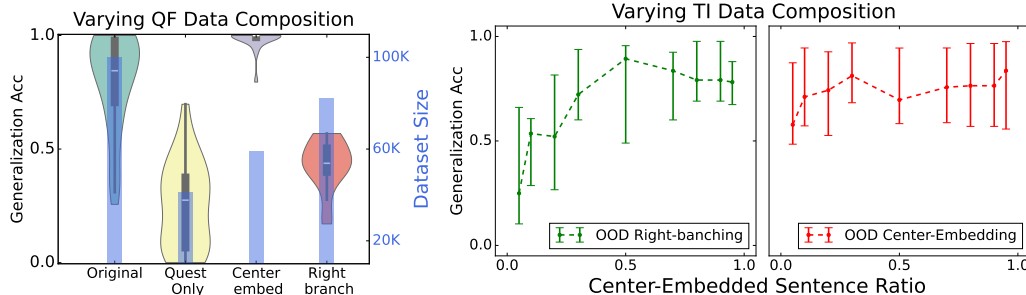

Figure 3: **Components of training data drive different generalization behaviors.** *Left:* Center-embedded sentences, which in the QF training data only appear in declaration copying examples, induce hierarchical generalization. *Right:* Models are trained on different data mixes and evaluated on two OOD sets: unambiguous right-branching sentences (*green*) and unambiguous center-embedded sentences (*red*). For center-embedded sentences, the hierarchical rule is preferred regardless of data mixes. For right-branching sentences, the model's preference for the hierarchical rule is exclusively driven by having a large mix of center-embedded sentences in the training data.

that the model needs to keep track of multiple subject-verb dependencies: one for the main clause and a separate one for the embedded relative clause. In these cases, modeling those subject-verb relationships with a tree structure is more compact and efficient.

## 4.2 QUESTION FORMATION

As specified in Section 3.1, the training data for QF must be ambiguous between the linear rule (i.e., moving the first auxiliary) and the hierarchical rule (i.e., moving the main auxiliary). Center-embedded sentences do not meet this ambiguity requirement and, therefore, cannot appear in question formation training samples. To ensure the model is exposed to diverse sentence types, McCoy et al. (2018) introduces a secondary task to the QF training dataset: declaration copying. Like question formation, the declaration-copying sample starts with a declarative sentence, but instead of transforming it, the model simply repeats it. Since the ambiguity requirement does not apply to the declaration-copying task, center-embedded sentences are included in this secondary task. Concrete examples of both tasks can be found in Appendix B.

We train models on three subsets of the original training data, varying the composition of the declaration-copying examples. In *Quest Only*, we remove all declaration copying examples. In *Center embed*, we only keep center-embedded examples. In *Right branch*, we only keep right-branching examples. For all three subsets, the question formation samples remain unchanged. Each setup reaches 100% in-distribution validation accuracy; however, the OOD generalization performance, shown in Figure 3 (*left*), differs significantly across the subsets. When the declaration-copying task is removed, none of the 50 runs achieve an OOD generalization accuracy above 75%, indicating that declaration copying examples are essential for inducing the hierarchical rule. When trained solely on center-embedded sentences in the declaration-copying task, models exhibit a strong preference for the hierarchical rule. In contrast, training only on right-branching sentences leads to poor hierarchical generalization. This evidence suggests that center-embedded sentences direct a model towards the hierarchical rule.

## 4.3 TENSE INFLECTION

We now analyze hierarchical generalization in the tense inflection task, demonstrating the generality of our findings across grammatical rules. Linzen et al. (2016) first proposed the idea to use a verb inflections to assess the model's grammatical capabilities. McCoy et al. (2020) then adopted the question formation data to the tense inflection task by creating the TI dataset using a set of CFG rules and vocabularies similar to the ones used for QF.

In the TI training data, both right-branching and center-embedded sentences are made ambiguous by ensuring the distractor noun shares the same plurality as the main subject. For right-branching sentences, since there isn't a relative clause modifying the subject, a preposition phrase modifying the subject provides the distractor noun. In contrast, for center-embedded sentences, since there is a

relative clause modifying the subject, either the subject or the object of the modifying clause can act as the distractor noun. We list examples below:

1. **Right Branching**: The noun in the prepositional phrase (e.g., " *to the cabinet*") acts as the distractor in the TI task.
   Example A (ID): *The keys to the **cabinet** are on the table.*
   Example B (OOD): *The keys to the **cabinets** are on the table.*
2. **Center Embedding**: Either the subject or the object inside the relative clause acts as the distractor in the TI task.
   Example C (ID): *The keys that unlock the **cabinets** are on the table.*
   Example D (OOD): *The keys that unlock the **cabinet** are on the table.*

We create variations of the TI training data by adjusting the ratio of right-branching to center-embedded sentences while keeping the total training size constant. The model is trained on nine different data mixes, and its generalization behavior is tested on two OOD sets: one containing unambiguous right-branching sentences (e.g., Example B) and the other containing unambiguous center-embedded sentences (e.g., Example D).

Generalization accuracies are shown in Figure 3 (*right*). When the training data is dominated by ambiguous right-branching sentences, the model fails to learn the hierarchical rule, as indicated by low OOD generalization accuracy on the left side of the green line. However, increasing the proportion of center-embedded sentences biases the model toward applying the hierarchical rule, even on right-branching sentences. This shift in behavior is reflected by improved generalization accuracy on the right side of the green line. The red line in Figure 3 (*right*) represents the model's generalization accuracy on unambiguous center-embedded sentences. Regardless of the data mix, the model consistently treats center embeddings as hierarchical and applies the hierarchical rule to OOD data. In contrast, the model only applies the hierarchical rule to right-branching sentences after being exposed to a sufficient quantity of center-embedded sentences during training. These observations suggest that center embeddings drive the model's overall preference for tree structures.

In Appendix C, we further partition center-embedded sentences based on the syntactic role of the main subject within the modifying clause. We show that while both subtypes induce the hierarchical rule in the QF task, one subtype provides a stronger hierarchical bias in the TI task.

## 5    TRAINING STABILIZES IF A MODEL COMMITS TO A RULE

Why do some runs fail to generalize hierarchically even when trained on hierarchical-inducing data? In this section, we will show that these failures are consequences of training instability; models only stabilize OOD if they commit to a general rule.

### 5.1    INSTABILITY DURING TRAINING

When training models on both QF and TI, some random seeds lead to highly unstable OOD behavior, with generalization accuracy often undergoing large swings during training. Furthermore, the unstable behavior is not *consistent* across different seeds. In Appendix F, we show examples of different OOD behaviors during training. We also show that both the instability and inconsistency in OOD behavior are significant only after the ID performance has converged. We measure instability across training time using *total variation* (TV). Specifically, we checkpoint the model every 2K steps and measure the generalization accuracy at each checkpoint, denoting as $\text{Acc}_i$. The total variation is defined as:

$$\text{Total Variation (TV)} = \frac{1}{|\text{ckpts}|} \sum_{i \in \text{ckpts}} |\text{Acc}_i - \text{Acc}_{i-1}|, \quad \text{where ckpts} = \{2K, 4K, 6K, \dots\}$$

### 5.2    TRAINING STABILITY TIES TO RULE COMMITMENT

We now demonstrate the connection between stable OOD behavior and rule commitment. We construct QF training datasets such that they contain different proportions of hierarchical-inducing (i.e., center-embedded) and linear-inducing (i.e., right-branching) declarations, while keeping questions constant. Further details on the dataset can be found in Appendix D.

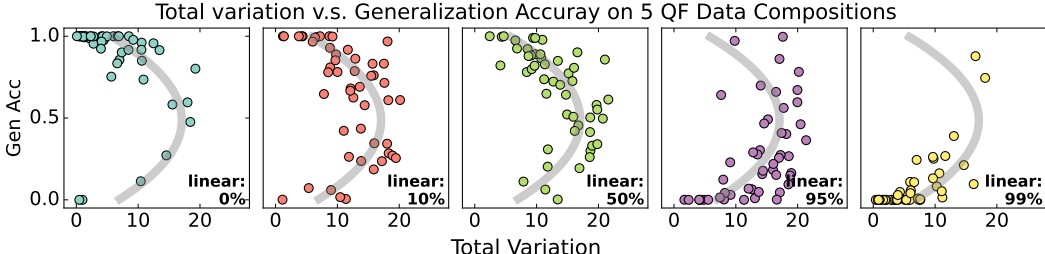

Figure 5: **Total variation across training v.s. final generalization accuracy for QF task.** OOD behavior stabilizes during training if a model commits to a simple rule. By mixing data that induces the linear and hierarchical rules, we can create conditions that allow models to stabilize in either rule. "Linear" denotes the proportion of linear-inducing declarations in the data. Grey line indicates the smoothed average curve across all runs and all five datasets.

Figure 4 shows the relationship between data homogeneity and training stability. When the training data is dominated by either linear-inducing (linear=99%) or hierarchy-inducing (linear=0%) examples, more random seeds lead to stable OOD curves. When the training data is a heterogeneous mix instead, potential rules compete, leading to unstable training. Figure 5 shows the relationship between training stability and generalization performance.

Across the five data mixes, the final generalization accuracy for all the stable models is either $100\%$ or $0\%$, indicating that the stable models have all committed to a general rule. While models can stabilize in either rule, data composition determines how likely a run is stable and for stable runes which rule is favored. Interestingly, when

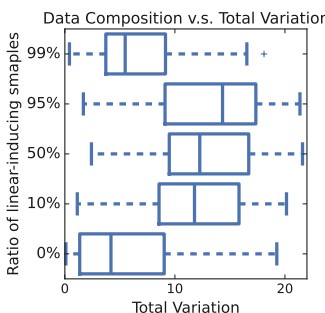

Figure 4: **Competition between subsets of data drives training instability.**

the data heterogeneous (e.g., linear=10% case), the final generalization accuracy for stable runs is bimodally distributed, clustering around $100\%$ or $0\%$. This bimodality suggests that training stability is always associated with a commitment to either the linear or the hierarchical rule, even when the data mix does not favor either rule.

In summary, with heterogeneous training data, competition between rules leads to more unstable training runs. Even with heterogeneous data mixes, however, some runs can still stabilize if they commit to one of the competing rules. In Appendix E.2, we replicate this analysis for the TI task, showing similar results.

## 6 DATA DIVERSITY LEADS TO GENERALIZATION

We have linked training stability to rule commitment. But why can't networks stabilize without committing to a rule? In this section, we will explore the non-monotonic relationship between data diversity, training instability, and rule commitment.

### 6.1 MEASURING DATA DIVERSITY

In order to measure the diversity of our training data, we must compute the syntactic similarity between different example sentences. We describe a sentence pair's similarity by the tree-edit distance (TED) of their latent tree representations (Chomsky, 2015). When two sentences share the same syntax tree, transforming one into the other requires only leaf-node (i.e., vocabulary) changes. For example, "*My unicorn entertains her tyrannosaurus*," and, "*Your zebra eats some apples*," have different vocabulary but identical syntax trees. We define data diversity as the number of unique syntactic trees in the training data. This way of using syntax TED to measure diversity data has been used in both natural language (Huang et al., 2023; Gao & He, 2024; Ramírez et al., 2022) and code (Song et al., 2024). We will show that when the model is exposed to a fewer unique syntax

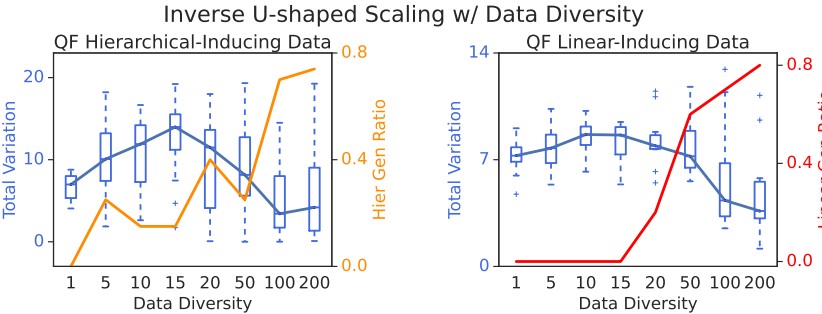

Figure 6: **Inverse U-shaped relationship between training stability and data diversity.** At low data diversity, training is stable but the model memorize individual syntactic patterns rather than committing to a rule. With moderate data diversity, training becomes unstable. As diversity increases further, the model commits to a rule and training stabilizes again.

trees during training, it memorizes those patterns without extrapolating any rules to unseen sentence structures. Consequently, the model fails to commit to a general rule.

## 6.2 INVERSE U-SHAPED SCALING

**Commitment to hierarchical rule**   We first control data diversity on datasets that induce hierarchical generalization in QF. We construct variations of the QF training data, each with 50K question samples and 50K hierarchical-inducing declarations, while varying the diversity of the declaration examples. We train 20 random seeds for each training set variation. To measure intra-run instability, we use total variation, and to assess hierarchical rule commitment, we report the proportion of runs achieving generalization accuracy >95%. Figure 6 (left) shows the distribution of total variation across 20 seeds and the corresponding hierarchical generalization ratios.

We observe an inverse U-shaped relationship between data diversity and training instability, revealing three distinct regimes. In the low-diversity regime, training is stable but the model fails to commit to a rule. In Appendix G, we further investigate this failure to rule commitment. We show that in trained on low diversity data, model can memorize specific syntax patterns and apply the hierarchical rule to to those structures but it cannot extrapolate the rule to unseen syntax structures. In the mid-diversity regime, training becomes unstable due to variation across batches. Overall, with insufficient diversity, relatively few runs can learn the hierarchical rule. Finally, in the high-diversity regime, training stabilizes again as the model commits to the hierarchical rule, indicated by a high hierarchical generalization ratio.

**Commitment to linear rule**   In Figure 5 right-most panel, the model has a strong preference to apply linear rule OOD when the training data contains 99% linear-inducing data (i.e., right branching sentences). However, Figure 3 (red violin) shows that when the training data contains *exclusively* linear-inducing sentences, models suddenly fail to apply the linear rule OOD either. We can use data diversity to explain the failure to rule commitment: right-branching sentences lack syntactic variation, as the main auxiliary always follows the subject noun. This lack of syntax diversity prevents rule extrapolation. By introducing as little as 1% of center-embedded sentences, we introduce the diversity necessary to consistent apply a rule OOD and the skewed ratio between the hierarchical and linear inducing sentences determine that the linear rule is preferred over the hierarchical rule.

To confirm that data diversity is also key to learning the linear rule, we create variations of QF training data with 50K questions and 50K declarations, including 99% right-branching and 1% center-embedded sentences. We control the diversity of *center-embedded* sentences as before and use the proportion of runs achieving generalization accuracy below 5% to quantify the likelihood of committing to the linear rule. As shown in Figure 6 (*right*), we observe a similar U-shaped scaling behavior, confirming that models only commit to a rule when trained on diverse data.

## 7   DISCUSSION AND CONCLUSIONS

By exploring the role of data structure in determining OOD generalization rules, we have also revealed which settings allow us to predict model behavior. We show that complex grammatical structures

guide models toward hierarchical rules, while mixed data compositions lead to unstable dynamics and inconsistent rule commitment. These findings emphasize the importance of understanding how data diversity shapes both stability and generalization in neural networks. Our findings have a number of implications across machine learning and even formal linguistics.

**Clusters of generalization behavior across seeds**    While errors are often treated as Gaussian noise in the theoretical literature, our findings suggest that errors may only be distributed unimodally for a given compositional solution. Our work joins the growing literature that suggests random variation not only has an effect, but can create clusters of OOD behaviors. Previously, clustered distributions have been documented in text classification heuristics (Juneja et al., 2022) and training dynamics (Hu et al., 2023). In our case, we note that generalization accuracy is only clearly multimodally distributed when specifically considering stable training runs. We suggest that research on compositional variation in training consider training stability in the future.

**Implications for formal linguistics**    Our findings have potential implications for linguistics debates about the poverty of the stimulus (McCoy et al., 2018; Berwick et al., 2011). Linguists have extensively studied the question of what data is necessary and sufficient to learn grammatical rules. In particular, Wexler (1980) argue that all English syntactic rules are learnable given "degree 2" data: sentences with only one embedded clause nested within another clause. Our mixed scoping results show that without a stronger architectural inductive bias—the very subject of the poverty of the stimulus debate—degree 1 data alone cannot induce a preference for hierarchical structure. However, our work also supports the position of Lightfoot (1989) that lower degree data is adequate for a child to learn a specific rule, as the LM generalizes ID degree 1 QF rule examples to OOD degree 2 by using the hierarchical inductive bias induced by declaration examples.

**Grokking, instability, and latent structure**    Murty et al. (2023), exploring the same data setting we do, call the transition from linear generalization to hierarchical generalization rules during training *structural grokking*. Classic grokking (Power et al., 2022), however, is different: Rather than a transition between generalization rules, it describes a transition from memorization to generalization.

Our findings clarify both scenarios. We link structural grokking to the instability formed by competition between linear- and hierarchical-inducing training subsets. Without competing subsets, the model immediately learns either the linear or the hierarchical rule without the gradual transition of structural grokking. This instability could represent the same phenomenon of circuit competition described by Ahuja et al. (2024). We find a similar pattern of instability in our study of data diversity, with implications for classic grokking. In this case, the competition is not between two rules, but instead between memorized heuristics—sufficient for modeling syntactically homogeneous training data—and simple OOD rules—required to efficiently model diverse training data. Yet again, while a strict memorization regime is relatively stable, the regime between memorization and generalization is unstable, leading to potential grokking.

Our findings suggest that memorization is just another rule that the model can adopt when it is the simplest way of capturing the training distribution. Such a framework unifies the grokking literature with other phenomena such as emergence (Schaeffer et al., 2023) and benign interpolation (Theunissen et al., 2020).

## ETHICS STATEMENT

This research does not present any direct ethical concerns. The work involves empirical studies of machine learning models and their behavior in language tasks. No human subjects, sensitive data, or high-stakes applications were involved in this research. Therefore, no specific ethical considerations were necessary for this work.

## REPRODUCIBILITY STATEMENT

All relevant details regarding the experimental setup including model architecture, hyperparameters, and data preprocessing, are included in the main text (Section 3.3) and appendices (Section D). Additionally, the code and scripts used to run the experiments are provided in the supplementary material and will be made publicly available upon acceptance.

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

# A    RELATED WORK EXTENDED

## A.1    SYNTAX AND HIERARCHICAL GENERALIZATION

While works mentioned in Section 2.1 focused on models trained from scratch, another line of research examined the inductive bias of pretrained models. Mueller et al. (2024); Mueller & Linzen (2023) pretrained transformers on text corpora such as Wikipedia and CHILDES (MacWhinney, 2014) before fine-tuning them on the question formation task. They found that exposure to large amounts of natural language data enables transformers to generalize hierarchically.

Instead of using the question formation task as a probe, Hewitt & Manning (2019); Murty et al. (2022) directly interpreted model's internal representation to understand whether transformers constrain their computations to to follow tree-structure patterns. Hewitt & Manning (2019) demonstrated that the syntax tress are embedded in model's representation space. Similarly, Murty et al. (2022) projects transformers into a tree-structured network, and showed that transformers become more tree-like over the course of training on language data.

## A.2    RANDOM VARIATION

Specific training choices, such as hyperparameters, are crucial to model outcomes. However, even when controlling for these factors, training machine learning models remains inherently stochastic—models can be sensitive to random initialization and the order of training examples. Zhou et al. (2020); D'Amour et al. (2022); Naik et al. (2018) reported significant performance differences across model checkpoints on various analysis and stress test sets. Zhou et al. (2020) further found that instability extends throughout the training curve, not just in final outcomes. To investigate the source of this inconsistency, Dodge et al. (2020) compared the effects of weight initialization and data order, concluding that both factors contribute equally to variations in out-of-sample performance.

Similarly, Sellam et al. (2021) found that repeating the pre-training process on BERT models can result in significantly different performances on downstream tasks. To promote more robust experimental testing, they introduced a set of 25 BERT-BASE checkpoints to ensure that experimental conclusions are not influenced by artifacts, such as specific instances of the model. In this work, we also observe training inconsistencies across runs on OOD data, both during training and at convergence. Unlike prior studies that focus on implications of random variations on experimental design, we study the source of training inconsistencies and link these inconsistencies to simplicity bias and the characteristics of the training data.

## A.3    SIMPLICITY BIAS

Models often favor simpler functions early in training, a phenomenon known as simplicity bias (Hermann & Lampinen, 2020), which is also common in LMs. Choshen et al. (2022) found that early LMs behave like n-gram models, and Saphra & Lopez (2019) observed that early LMs learn simplified versions of the language modeling task. McCoy et al. (2019) showed that even fully trained models can rely on simple heuristics, like lexical overlap, to perform well on Natural Language Inference (NLI) tasks. Chen et al. (2023) further explored the connection between training dynamics and simplicity bias, showing that simpler functions learned early on can continue to influence fully trained models, and mitigating this bias can have long-term effects on training outcomes.

Phase transitions have been identified as markers of shifts from simplistic heuristics to more complex model behavior, often triggered by the amount of training data or model size. In language models, Olsson et al. (2022) showed that the emergence of induction heads in autoregressive models is linked to handling longer context sizes and in-context learning. Similar phase transitions have been studied in non-language domains, such as algorithmic tasks (Power et al., 2022; Merrill et al., 2023) and arithmetic tasks (Nanda et al., 2023; Barak et al., 2022).

In the context of hierarchical generalization, Ahuja et al. (2024) used a Bayesian approach to analyze the simplicity of hierarchical versus linear rules in modeling English syntax. They argued that transformers favor the hierarchical rule because it is simpler than the linear rule. However, their model fails to explain (1) why learning the hierarchical rule is delayed (i.e., after learning the linear rule) and (2) why hierarchical generalization is inconsistent across runs. In this work, we offer a different perspective, showing that a model's simplicity bias towards either rule is driven by the characteristics of the training data.

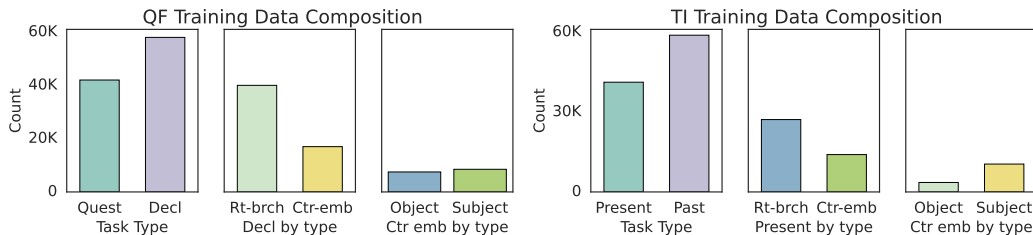

Figure 7: **Components of the original QF and TI training data.** *Left:* QF training data contains samples of two tasks types: question formation and declaration copying. We further break down samples in the declaration copying task by branching type. We also breakdown center-embedded sentences based on whether the main subject serves the subject or object in the embedded clause. *Right:* TI training data also contains samples of two task types: tense inflection and past tense copying. Similar to QF, we further breakdown tense inflection samples by branching types, and center-embedded sentences (in the tense inflection samples) by subject/object type.

## B TRAINING DATA SAMPLES

### B.1 QUESTION FORMATION

When we mention "declarations," we are referring to the declaration copying task, and "questions" refer to the question formation task. Here are two examples randomly taken from the training data:

- Declaration Example: `our zebra doesn't applaud the unicorn .  decl our zebra doesn't applaud the unicorn .`
- Question Example: `some unicorns do move .  quest do some unicorns move ?`

Both tasks begin with an input declarative sentence, followed by a task indicator token (`decl` or `quest`), and end with the output. During training, the entire sequence is used in the causal language modeling objective. The in-distribution validation set and the OOD generalization set only contain question formation samples.

### B.2 TENSE INFLECTION

- Past Example: `our peacocks above our walruses amused your zebras . PAST our peacocks above our walruses amused your zebras .`
- Present Example: `your unicorns that our xylophones comforted swam . PRESENT your unicorns that our xylophones comfort swim .`

The tense inflection task is indicate by the `PRESENT` token, and in Section 4.3, we only used tense inflection samples during training. In Appendix E.1, we further explore the use of a secondary copying task to achieve OOD generalization. Similar to the question formation training data, the secondary task only requires repeating the given sentence, which is always in the past tense, and the copying task is marked by the `PRESENT` token.

## C FURTHER PARTITIONS ON CENTER-EMBEDDED SENTENCES

### C.1 TWO SUBTYPES OF CENTER-EMBEDDED SENTENCES

In Section 4, we showed that center-embedded sentences drive hierarchical generalization in both the QF and TI tasks. Here, we further partition center-embedded sentences based on the syntactic role of the *main subject* (i.e., the subject of the main clause) within the modifying clause. Specifically, we classify them into two types:

1. **Subject-type**: The main subject serves as the **subject** within the clause.
   Example: *The keys that unlock the cabinet are on the table.*
2. **Object-type**: The main subject serves as the **object** within the clause.
   Example: *The keys that the bear uses are on the table.*

This partition is motivated by their distinct subject-verb dependency patterns. In subject-type sentences, both the main verb (from the main clause) and the embedded verb (from the relative clause) depend on the main subject. In contrast, object-type sentences exhibit a nested subject-verb structure. Our goal is to investigate whether differences in subject-verb dependency patterns influence the model's preference for the hierarchical rule.

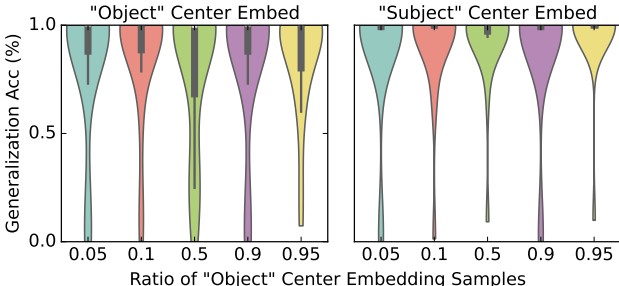

Figure 8: **Both subtypes of center-embedded sentences induces hierarchical generalization in QF.** We train models on datasets containing different ratios of object-type v.s. subject-type center-embedded sentences. We then evaluate on models on two OOD generalization set, one containing unambiguous object-type center-embedded sentences and the other unambiguous subject-type center-embedded sentences.

## C.2 QF TASK

We first investigate whether the two subtypes of center-embedded sentences differentially influence the model's preference for the hierarchical rule in the QF task. For all training data variants, we fix 50K question formation samples and 50K declaration copying samples, with the latter containing only center-embedded sentences but varying the ratio between the two subtypes. To analyze generalization behavior on a more granular level, we partition the generalization set (composed solely of center-embedded sentences) into the two subtypes as well. Models are trained on 30 random seeds, and results are shown in Figure 8. Regardless of the data mix, the model consistently favors the hierarchical rule across both partitions of the generalization set. This suggests that, for question formation, both subtypes of center-embedded sentences equally contribute to the model's ability to identify the main auxiliary.

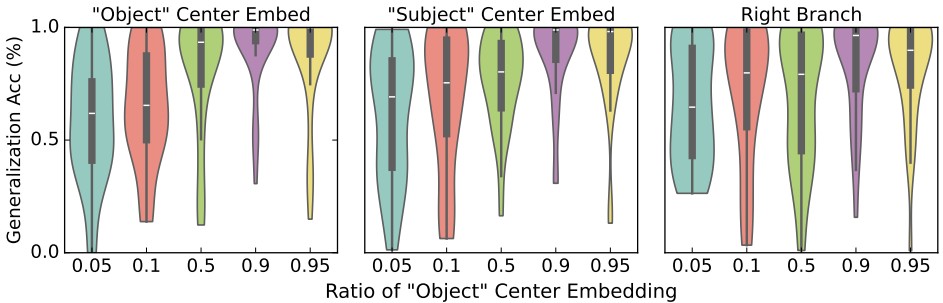

Figure 9: **Object-type center-embedded sentences gives a stronger bias towards hierarchical generalization in TI.** We train models on datasets containing different ratios of object-type v.s. subject-type center-embedded sentences. We then evaluate on models on three OOD generalization set, one containing unambiguous object-type center-embedded sentences, one unambiguous subject-type center-embedded sentences, and one unambiguous right-branching sentences.

## C.3 TI TASK

We repeat a similar experiment for the TI task, fixing the total number of tense inflection samples to 100K. As shown in Section 4.3, models exhibit the strongest hierarchical generalization when trained on primarily center-embedded sentences. Therefore, in the following data variants, 99% of the samples are center-embedded sentences, with the remaining 1% being right-branching sentences. Within the center-embedded samples, we vary the ratio between the two subtypes. To evaluate

generalization, we split the generalization set into three groups: the two subtypes of center-embedded sentences and right-branching sentences. Models trained on 30 random seeds show that, across all three generalization sets, accuracy is positively correlated with the proportion of object-type center-embedded sentences (Figure 9). However, even when models are trained predominantly on subject-type center-embedded sentences (teal violins in Figure 9), they still show a strong tendency toward hierarchical generalization. Thus, while both subtypes drive hierarchical generalization in TI, object-type center-embedded sentences have a stronger effect. Notably, the original TI training data includes a higher proportion of right-branching sentences (shown in 7) and a higher ratio of subject-type center-embedded sentences—both of which are suboptimal for inducing hierarchical generalization.

## D    VARYING DATA RATIOS FOR QUESTION FORMATION

**Data composition details**    We construct variations of the training data using the following procedure. Each new dataset contains 50K questions (reused from the original data) and 50K declarations, where we control the ratio between center-embedded and right-branching sentences. These datasets are used for the experiments in Section 5.2. To generate additional declarations, we keep the distribution of the unique syntax structures in original dataset. Specifically, for each sentence in the original data, we extract the syntax tree using the CGF rules and resample words from the vocabulary to create new sentence samples.

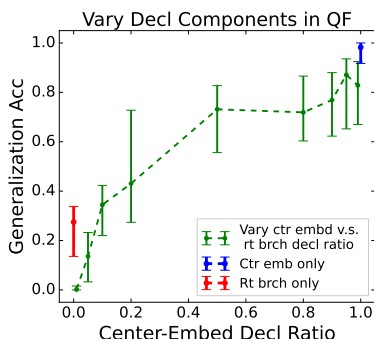

Figure 10: **Hierarchical generalization in QF is sensitive to compositions of declaration-copying samples.**

**Sensitivity to data compositions**    We use the five datasets above to examine how different mix ratios affect a model's preference towards the hierarchical generalization. The median generalization accuracy, along with error bars representing the 35th and 65th percentiles, is shown in Figure 10. First, note that there is a sharp performance drop between the blue bar and the right-most green bar. This sharp transition indicates that mixing in as little as 1% of right-branching declarations significantly reduces the model's likelihood of generalizing hierarchically. Interestingly, when the dataset is predominantly right-branching declarations, models consistently achieve 0% generalization accuracy, indicating a strong preference for the linear rule across all training runs. However, note that there is another sharp transition between the red bard and the left-most green bar. This transition indicates that as soon as we remove the 1% of center-embedded sentences, the model fails to learn either the linear rule or the hierarchical rule. As a result, the generalization accuracy is close to random guess ($\sim 25\%$). This transition is closely studied in Section 6.1, where we examine how data diversity leads to rule commitment.

## E    ADDITIONAL RESULTS ON TENSE INFLECTION

### E.1    A SECONDARY TASK IS NOT NECESSARY

In the original of TI training data (McCoy et al., 2020), a secondary task is also included to mimic the question formation training data. In this secondary task, instead of transforming a sentence from the past tense to the present tense, the model simply needs to repeat it. For concrete examples, see Appendix B. Figure 7 (*right*) shows a breakdown of the two tasks in the original TI training data. In experiments conducted in Section 3.2, we have eliminated the used of this secondary task because center-embedded sentences can be included in the tense inflection training samples *without* violating the ambiguity requirement. Here, we use the training data originally proposed by Mc-Coy et al. (2020) to confirm that the use of secondary task is indeed not necessary. Specifically, we remove all the past-tense-copying samples from the original training data and train models on the tense-inflection task only. We

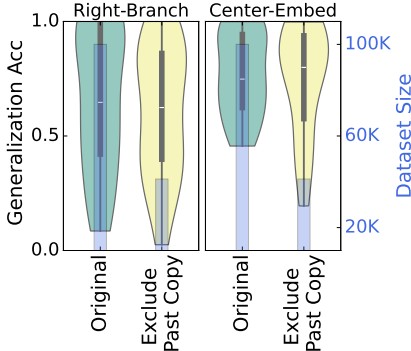

Figure 11: **Past-copy task is not necessary to induce hierarchical generalization in TI.**

evaluate the model's generalization performance on two OOD set containing unambiguous right-branching and unambiguous center-embedded sentences, shown in Figure 11. We can see that the model's OOD performances are the same with or without the secondary task.

### E.2 TRAINING INSTABILITY AND RULE COMMITMENT FOR TENSE INFLECTION

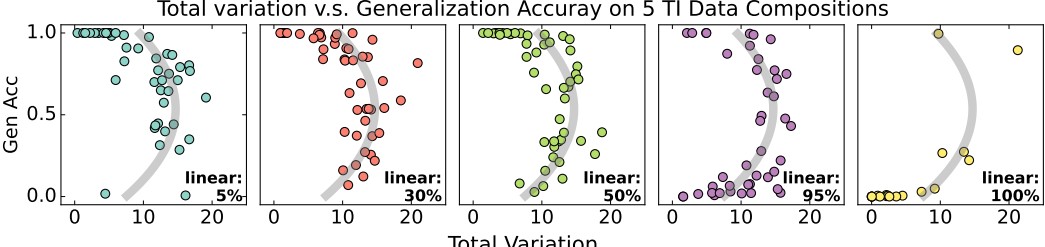

Figure 12: **Total Variation v.s. final generalization accuracy for TI task.** Similar to Figure 5, we observe the same horseshoe shaped behavior between training stability and final generalization accuracy on right-branching sentences for the TI task.

We repeat the same total variation analysis in Section 5 for the tense inflection task. We use the data mixes from Section 4.3. Specifically, we include only tense inflection samples and vary the ratio between linear-inducing (i.e., right-branching) and hierarchical-inducing (i.e., center-embedded) sentences. In Section 4.3, we have already concluded that the hierarchical rule is *always* preferred for center-embedded sentences regardless of data mixes. For this reason, we are interested in examining the rule preference and training stability for unambiguous right-branching sentences. In Figure 12 we visualize the relationship between total variation and the final generalization accuracy on unambiguous right-branching sentences. The qualitative behavior is similar to what we have observed in QF (Section 5.2).

## F TRAINING INSTABILITY

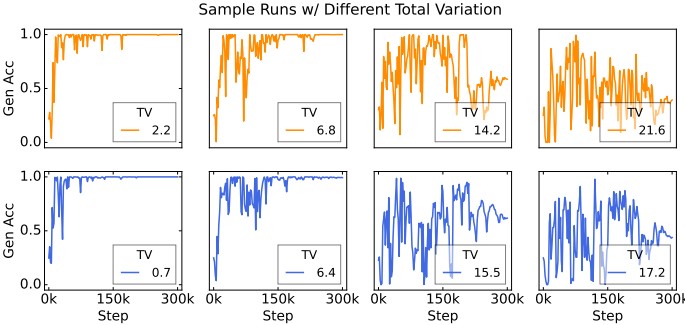

Figure 13: **Each training run either stabilizes in a simple OOD generalization rule or oscillates in its OOD accuracy.** The OOD generalization behaviors can be either stable or unstable when trained on different seeds. We use total variation to quantify the instability within one training run.

In Figure 14, we visualize the training dynamics for 30 independent runs when trained on the original QF data. Each run differs in both model initialization and data order. Notice that the training dynamics for runs exhibit grokking behaviors: OOD generalization is delayed when compared to training loss convergence and validation performance convergence. These runs share a similar progression in training loss, validation accuracy, and generalization accuracy up until moment when the training loss converges. Interestingly, after convergence on training loss, all runs reach $0\%$ on the generalization set, indicating that the model strictly prefers linear rules on OOD data. After that, models start to achieve non-trivial performance in generalization accuracy. However, for many runs the generalization accuracy does not increase monotonically. Instead, we observe massive swings in generalization accuracy during this training period as well as large inconsistency across different seeds. Overall, training is *always* stable for ID data while the performance for OOD data is inconsistent across seeds. We visualize runs with different of total variation values in Figure 13.

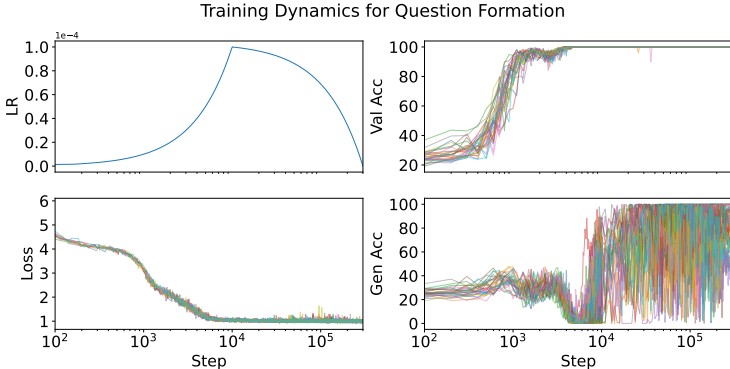

Figure 14: **Training Dynamics on original question formation data.** Training loss and in-distribution validation accuracy is stable during training and consistent across random seeds. In contrast, the model's performance on OOD data is both unstable during training and inconsistent across seeds. The instability and inconsistency is most prominent during grokking (i.e., when training loss has converged).

# G  DATA DIVERSITY AND MEMORIZATION PATTERNS

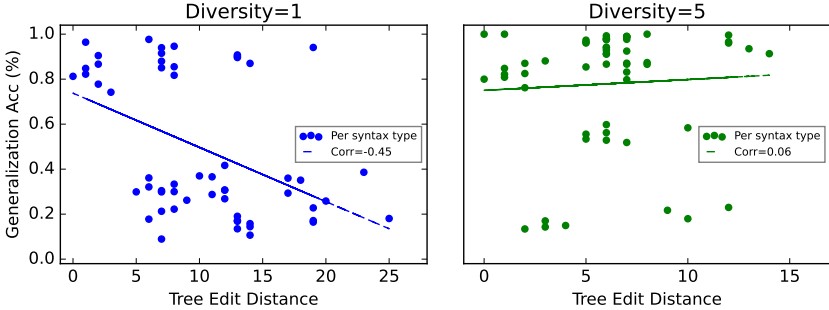

Figure 15: **OOD generalization v.s. syntax similarity to training data.** At low data diversity, model memorizes syntax patterns and applies the hierarchical rule only syntax structures similar to ones in the training data. With higher data diversity, model extrapolates rules and can apply the hierarchical rule even to unseen syntax structures that are dissimilar to training data.

We investigate model behavior when trained on data with limited diversity. By analyzing a model's generalization accuracy across different syntactic types, we aim to distinguish patterns indicative of either memorization or generalization.

**Measuring data similarity**    Building on the diversity measure from Section 6.1, we now use Tree-Edit Distance (TED) as a measure of sentence similarity. As before, we first construct syntax trees using CFG rules, then calculate TED using the Zhang-Shasha Tree-Edit Distance algorithm (Zhang & Shasha, 1989). We define TED=0 for sentences that share the same syntax structure but differ only in vocabulary. This similarity measure allows us to quantify, for each sample in the OOD generalization set, the closest matching sentence type in the training data. In the memorization regime, where the model encounters only a few syntax types, we suspect it cannot extrapolate rules to syntactically distinct OOD sentences. In contrast, with a more diverse syntax exposure, rule extrapolation may enable the model to apply rules even to OOD sentence types.

**Experiment**    To verify our intuition about memorization and generalization, we train models on two variations of the QF data. In the first variation, the declaration-copying task has data diversity set to 1, meaning only one syntax type appears, and we specifically choose one with center embedding. In the second variation, the declaration-copying task has diversity set to 5, with all 5 types containing center embeddings. For both datasets, the question-formation task remains unchanged, consisting solely of right-branching sentences. For the diversity=1 dataset, we calculate TED for each unique

syntax type in the OOD set against the single syntax type in the declaration-copying task. For the diversity=5 dataset, we compute TED between each OOD sample and the five syntax types in the declaration-copying task, taking the minimum. This TED score provides a measure of similarity between the OOD samples and those encountered during training. Our goal is to determine, based on training with these datasets, which OOD syntax types the model applies the hierarchical rule to.

**Result** In Figure 15, we visualize the final generalization accuracy for each OOD syntax type against its TED relative to the training data. When trained on low-diversity data (Figure 15, *left*), generalization accuracy is negatively correlated with TED. For syntax types seen in the declaration-copying task (TED=0) and those similar to it, the model applies the hierarchical rule. However, for syntax types with high TED, the model's behavior is random (25%), indicating failure to follow any rule. As data diversity increases slightly (Figure 15, *right*), generalization accuracy no longer correlates with TED, suggesting that once the model begins to extrapolate the hierarchical rule, it can apply this rule to a wider range of OOD syntax types.

