# OpenReview forum: "Sometimes I am a Tree: Data Drives Unstable Hierarchical Generalization"
_ICLR.cc/2025/Conference — Submitted to ICLR 2025_

### Official Review · Reviewer_Sgqh · 2024-10-27

**Soundness:** 2
**Presentation:** 3
**Contribution:** 1
**Rating:** 3
**Confidence:** 3

**Summary:**

This paper investigate generalization of LMs.  Specilically, data composition significantly influences a model's out-of-distribution (OOD) generalization.  Insufficient data diversity can lead models to rely on memorization rather than achieving true generalization. They primarily validate these observations through learning on grammatical tasks.

model: 12M  decoder-only Transformer.
tasks: question formation, tense inflection.

**Strengths:**

The paper is well-written, with comprehensive visuals and thorough experimental results.

**Weaknesses:**

1. The conclusion in this paper have many overlaps with previous studies[1],[2],[3]; please specifiy your difference and contribution detailed.
2. The experiments only contain two dataset: question formation and tense inflection. This limited validation is insufficient to support the overall conclusions of the paper, generally speaking.  More tasks and models are needed.  Suggest tasks: language modeling, mathematics reasoning … …
3. The authors present only the experimental observation that "lower diversity data tends to promote memorization, while higher diversity data encourages generalization." These are  obvious conclusions compared with previous papers. Are there any new insights offering deeper theoretical reasoning or more controlled experiments to support these findings? or fuerther verfiication on LLMs, e.g. 7B ?

### Grammar

Line 113: “. . “

Line 119: multiple abbreviation definitions

Table 1: revise “Left” and “right” to “Top” and “Bottom”

[1] Liu Z, Michaud E J, Tegmark M. Omnigrok: Grokking beyond algorithmic data[C]//The Eleventh International Conference on Learning Representations. 2022.

[2] Zhu X, Fu Y, Zhou B, et al. Critical data size of language models from a grokking perspective[J]. arXiv preprint arXiv:2401.10463, 2024.

[3] Wang B, Yue X, Su Y, et al. Grokked Transformers are Implicit Reasoners: A Mechanistic Journey to the Edge of Generalization[J]. arXiv preprint arXiv:2405.15071, 2024.

**Questions:**

See in Weaknesses.

---

> ### Author Response · Authors · 2024-11-16
> **Reply to Sgqh's comment #1**
>
> Thank you for your valuable feedback! We provide our response below:
>
> > **Weakness 1:** The conclusion in this paper have many overlaps with previous studies[1],[2],[3]; please specifiy your difference and contribution detailed.
>
> Thank you for these references. Our paper could benefit from a more clear comparison with existing work related to grokking, which we have added. Unlike [1,2], we are not focused exclusively on competition between memorization and generalization, but also on competition between different generalization rules.  Importantly, we characterize the unstable regime in both data diversity and data complexity and we address connections between training stability and consistency under random variation. None of these phenomena are documented in the existing literature on grokking.   We discuss some of the connections with our work below.
>
> - Omnigrok [1]: this paper studies how grokking can be induced with different weight norms, associating generalization with a specific goldilocks zone weight norm value. While we do study memorization/generalization “classic” grokking transitions, we also study the transition between rules (i.e., structural grokking). While [1] focuses on weight complexity, we focus on data complexity. A possible question would be whether data diversity and complexity drive the weights towards an existing goldilocks zone, or whether they adjust the location of that goldilocks zone to be more accessible. Unifying model and corpus complexity would be an interesting direction for future work.
>
> - Critical data size [2]: this paper shows generalization can only happen when data is large enough. We might regard dataset size as a proxy for data diversity, and we do find that higher diversity leads to more rule-based generalization, but the presence of multiple possible generalization rules introduces another source of complexity in our setting. Furthermore, diversity in natural language data can be described with respect to syntax or other properties, not just dataset size.
>
> - Grokked transformers are implicit reasoners [3]: this paper studies OOD generalization in compositional reasoning. This paper found that it is not the data size that matters but specific aspects of data (in their case inferred/atomic ratio). Their finding could arguably be described as demonstrating the importance of another type of complexity, as we illustrate in the case of center embeddings inducing hierarchical structure (Section 4). Both [3] and our work look at model behavior during training, although we study stability while they study internal circuit formation. Our paper unifies [3] (complexity induces generalization) and [2] (diversity induces generalization) by highlighting that the latter promotes rules over memorization, while the former promotes complex rules over simple shortcut rules.
>
> For a clarification of our core contributions and broader framework, please refer to our top level rebuttal comment, which summarizes our findings and how they relate to each other. We specifically acknowledge that the broad relationship between diversity and generalization, and the relationship between complexity and rule selection, are both extensions of existing views to a new setting. However, all of our other findings are entirely novel, as is the disentanglement of diversity and complexity in promoting generalization.

---

> ### Author Response · Authors · 2024-11-16
> **Reply to Sgqh's comment #2**
>
> > **Weakness 2:**  “The experiments only contain two dataset: question formation and tense inflection. This limited validation is insufficient to support the overall conclusions of the paper, generally speaking. More tasks and models are needed. Suggest tasks: language modeling, mathematics reasoning … …”
>
> Unfortunately, there are few settings that allow systematic analysis of OOD rule selection, as these tasks do. We may be able to construct synthetic settings, but grammar learning has the advantage of being relatively well understood, with an existing literature to draw on.
>
> - Regarding language modeling:
>
>   The two case studies (QF and TI) are canonical subtasks for language modeling. This setting is actually a deconstructed language modeling task; a full language modeling task is too complex to isolate and analyze any one generalization rule at a reasonable scale, but it is likely that LLMs exhibit something like this behavior in cases of emergence (i.e., grokking on compositional tasks) and we will highlight that connection as an avenue for future work.
>
> - Regarding arithmetic reasoning:
>
>   In Section 5, we have shown that the grokking when generalization rules compete (i.e.,  “structural grokking”) is more nuanced than grokking in arithmetic tasks. Arithmetic tasks only reflect the memorization vs generalization axis, which we find to be governed by syntax diversity in English grammar; in arithmetic, this axis is governed by dataset size (ie, sample diversity). Existing results, such as the reviewer has cited [2], do find that grokking only occurs when 70% of the data is included. Interestingly, the training dynamics in arithmetic also cluster by random seed, according to Hu et al. [4], into 3 clusters---seemingly into highly unstable/memorization/generalization categories.
>
> These connections are interesting and certainly support our framework’s predictions relating homogeneity to memorization, but unfortunately arithmetic does not provide the second source of competition: rule selection. Therefore, it cannot incorporate a second axis of data complexity.

---

> > ### Author Response · Authors · 2024-11-16
> > **Reply to Sgqh's comment #3**
> >
> > > **Weakness 3:**  “The authors present only the experimental observation that "lower diversity data tends to promote memorization, while higher diversity data encourages generalization." These are obvious conclusions compared with previous papers. Are there any new insights offering deeper theoretical reasoning or more controlled experiments to support these findings? or fuerther verfiication on LLMs, e.g. 7B ?”
> >
> > As you have pointed out, in existing grokking literature, only the role of dataset size on grokking has been studied. We are not aware of any work that emphasizes the data diversity in arithmetic grokking or structural grokking, although some existing work connects diversity or complexity to generalization broadly, e.g., Berlot-Attwell et al. [5].
> >
> > To confirm the connection to grokking, in Appendix F we have added additional experiments, showing models trained on low diversity data will memorize syntactic patterns, whereas models trained on high diversity data will extrapolate rules to unseen syntax patterns. This experiment more closely reflects the memorization-to-generalization transition in classic grokking work. However, our paper also addresses the OOD rule competition (Section 5) discussed in more recent *structural* grokking work. We have restructured our paper and expanded the literature review to emphasize these connections.
> >
> > Furthermore, data diversity is not the only focus of our paper. We explore data on two axes, diversity and complexity. We find that diversity promotes rule-based generalization over memorization, whereas complexity promotes complex rules over simple rules.
> > We also highlight our findings about the unstable regime of OOD training dynamics and the inconsistent regime of random variation (Section 5), neither of which are present in the existing literature. Furthermore, we connect training instability with random variability through the horseshoe curves in Fig. 4, which demonstrate that data conditions can move the probability distribution closer to one rule or the other, but in doing so they also change the probability of unstable behavior predictably.
> >
> > **References:**
> >
> > [1] Liu, Ziming, Eric J. Michaud, and Max Tegmark. "Omnigrok: Grokking beyond algorithmic data." The Eleventh International Conference on Learning Representations. 2022.
> >
> > [2] Zhu, Xuekai, et al. "Critical data size of language models from a grokking perspective." arXiv preprint arXiv:2401.10463 (2024).
> >
> > [3] Wang, Boshi, et al. "Grokked Transformers are Implicit Reasoners: A Mechanistic Journey to the Edge of Generalization." arXiv preprint arXiv:2405.15071 (2024).
> >
> > [4] Hu, Michael Y., et al. "Latent state models of training dynamics." arXiv preprint arXiv:2308.09543 (2023).
> >
> > [5] Berlot-Attwell, Ian, et al. "Attribute Diversity Determines the Systematicity Gap in VQA." arXiv preprint arXiv:2311.08695 (2023).

---

> ### Author Response · Authors · 2024-11-25
> **Followup w/ Sgqh**
>
> We wanted to kindly follow up on our responses. If you have any further thoughts or clarifications on the points we have addressed, we would greatly appreciate your feedback! Please let us know if there’s anything specific you’d like me to elaborate on further. Thank you!

---

> > ### Comment · Reviewer_Sgqh · 2024-11-27
> >
> > Thank you for your detailed rebuttal and clarification.
> >
> > Partially my misunderstanding is corrected. However, the most important points are (1) more verification of real-world tasks and deeper theoretical investigation.  If you fix these problems, I think it might be a great work.
> > **However, I think this paper has not been accomplished (for now). So I will keep the score as it is.**
> >
> > Thank you once again for your rebuttal.

---

### Official Review · Reviewer_4ELk · 2024-10-28

**Soundness:** 4
**Presentation:** 3
**Contribution:** 4
**Rating:** 8
**Confidence:** 4

**Summary:**

This paper investigates the influence of training data composition on the generalization behaviors of language models, focusing on their ability to learn hierarchical syntactic representations versus relying on surface-level heuristics such as n-gram models. Through case studies involving English grammar tasks (specifically question formation and tense inflection) the authors show that the complexity and diversity of training data play pivotal roles in determining if models adopt hierarchical rules or simpler linear heuristics.
Interesting findings of the paper:
i) Adding sentences with deep syntactic trees in the training data encourages models to develop hierarchical syntactic representations  (and this seems to enable out of distribution generalization).
ii) When the training dataset have a mix of simple and complex grammatical structures, models show unstable training dynamics and inconsistent rule commitments across different random seeds (this aligns with findings by McCoy et al. (2018, 2020))
iii) In case of low data diversity, models tend to memorize patterns without learning robust hierarchical or linear rules, resulting in poor generalization.

I think the main message of this paper is the relevance of training data features in shaping the inductive biases of neural networks.

**Strengths:**

- Interesting analysis of the mechanisms of rule commitment.
- The identification of a memorization regime, where models stabilize without learning either hierarchical or linear rules.
- The findings are replicated across two distinct grammatical tasks, and backed up by linguistics theroies.

**Weaknesses:**

- The experiments utilize relatively small transformer models trained on synthetic datasets with 100K samples (it doesn't make the study less valid, but larger models may exhibit different inductive biases and learning dynamics that are not captured by smaller-scale experiments).
- The authors use a fixed set of hyperparameters (learning rate of 1e-4, Adam optimizer, specific layer configurations) across all experiments, without addressing how variations in hyperparameters might influence the model's ability to learn hierarchical rules or affect training stability.

**Questions:**

- It might be worthy study how other objectives (like MLM) interact with data composition to affect rule learning.
- Have you considered how the role of data composition might vary with languages that have different syntactic structures?

---

> ### Author Response · Authors · 2024-11-16
> **Reply to 4ELk**
>
> Thank you for your valuable feedback! We are glad that you find our work interesting. We provide our response below.
>
> ### **Weakness 1**
>
> You are entirely correct that this is something of a toy setting. Much of our results are engaged with the literature on grokking and random variation, and relative to that literature we are arguably far more realistic than most work, so thank you for pointing out that the study is still valid even if the inductive biases might change at different scales.
>
> ### **Weakness 2**
>
> Similar to grokking, the exact condition to induce structural grokking  is precarious. The early viability of memorization (for grokking) or linear rules (for structural grokking) is highly sensitive to hyperparameter settings, as has been widely acknowledged in the grokking literature. We agree that future work could address how hyperparameters such as batch size interact with data composition, given how precarious the clusters in generalization behavior are
>
> ### **Question 1**
>
> Given the findings of Ahuja et al. [1] that the language modeling objective is key to hierarchical inductive bias in general, we agree that the interaction between these data factors and the objective would be a rich area for future work.
>
> ### **Question 2**
> We are excited about the prospect of followup work that is more focused on the computational linguistics implications of our findings, rather than on training dynamics and random variation. Thank you for highlighting other languages, which we agree would be key (along with additional tasks)!
>
> [1] Ahuja, Kabir, et al. "Learning Syntax Without Planting Trees: Understanding When and Why Transformers Generalize Hierarchically." arXiv preprint arXiv:2404.16367 (2024).

---

> ### Comment · Reviewer_4ELk · 2024-11-20
>
> Thank you for you reply.
> I noticed that among the reviewer I am the one that gave this paper the highest score: I read the other reviews, and unless there is a major problem with the paper that we all overlooked, i'd like to retain my score of 8.
> I understand what my colleagues reviewer pointed out, but I still think that this paper is interesting.

---

### Official Review · Reviewer_tfb9 · 2024-10-31

**Soundness:** 3
**Presentation:** 2
**Contribution:** 2
**Rating:** 5
**Confidence:** 4

**Summary:**

This study analyzes the role of the syntactic complexity in language models’ acquisition of hierarchical inductive biases. Specifically, by generating sentences with varying syntax tree depths and of varying structures (some with mixed scoping and others with forward scoping), the features that lead to linear or hierarchical generalization are isolated. Additionally, by mixing these types of data, the authors conduct a grokking-style analysis of when stable, unstable, hierarchical, linear, or mixed generalizations are likely to be learned.

It is found that including sentences with a syntactic tree depth of at least 3 is required for hierarchical generalizations to be stably acquired, and that mixing depths in the training data can lead to unstable generalizations. It is also found that if the data is not sufficiently diverse, memorization (rather than any consistent generalization) is the preferred strategy.

**Strengths:**

* The paper adds to a rich literature on directly evaluating the inductive biases of language models. This paper approaches it from a grokking perspective, which, to my knowledge, has not been done before.
* In Fig. 4, it is interesting that such a small variety of syntactic structures in the declaratives would lead to hierarchical generalization. I had assumed that a more naturalistic and varied distribution would be necessary to prevent memorization.
* Thorough analysis of variation across many random seeds.

**Weaknesses:**

1. The findings could be much better contextualized with related work throughout the paper. For example, at L150: this is not the first work to analyze the effect of data on syntactic generalization. Mueller et al. (2023) (who are currently cited only in the Appendix) do a similar analysis, finding that simpler corpora lead to syntactic generalization with less data. Additionally, when referring to the syntactic transformations task setup at L144, Frank & Mathis (2007) [1] is the correct citation. At L154-161 (and App. F): it would be nice to contextualize this with analyses from McCoy et al. (2020) on instability across random seeds. In Sec. 4.3, these findings are very related to those of Papadimitriou & Jurafsky (2023) [2] (who are not cited) on training with recursive and/or cross-serial dependencies.
2. The findings are somewhat obvious in light of the above citations, and in light of grokking work (which, by the way, should also be better cited and discussed; see [3,4,5]). Data determines generalization because neural networks are a statistical approach to learning. There are probably new insights to be gleaned from this new task setting compared to past grokking work, and it could be nice to explicitly enumerate these in the paper.
3. L212-215: It is not clear why these hyperparameters were chosen, and why different hyperparameters were used for the two tasks.
4. Related to the first point (but more minor), the decision of what to include in the main paper vs. the appendix Related Work sections currently feels arbitrary.

**Questions:**

Questions
===
1. Where did the hyperparameters in L212-215 come from? Was there some tuning involved, and if so, would it be possible to show results for other tested hyperparameters?
2. By syntactic complexity, do you generally mean syntactic tree depth? The rarity/difficulty for humans in processing of particular structures? Number of edges in the tree (which could correlate with sentence length, even if the maximum depth of the tree were low)? I assume the first given the analyses in Sec. 4, but would be nice to explicitly define the term.
3. L420: which studies?
4. At first glance, Fig. 4 and Sec. 6 feel contradictory. There is some interesting nuance here that could be interesting to explore: diversity is necessary to prevent memorization, but at the same time, including a diverse mixture of complex and simple sentences leads to *worse* generalization than simply using more complex sentences. Why is this?

Suggestions/Typos
===
* L182: there are *at least* two strategies. The model could rely on other heuristics, such as always moving the affirmative verb in QF as opposed to the main or first verb (as attested in Mueller et al., 2024).
* Table 1 caption: by “left” and “right”, do you mean “top” and “bottom”?
* L113: two periods

References
===
[1] Robert Frank & Donald Mathis (2007). “Transformational networks.” Models of Human Language Acquisition. https://bpb-us-e2.wpmucdn.com/websites.umass.edu/dist/a/27637/files/2017/06/cogsci-2007.pdf

[2] Isabel Papadimitriou & Dan Jurafsky (2023). “Injecting structural hints: Using language models to study inductive biases in language learning.” Findings of EMNLP. https://aclanthology.org/2023.findings-emnlp.563/

[3] Yifei Huang et al. (2024). “Unified View of Grokking, Double Descent and Emergent Abilities: A Perspective from Circuits Competition.” COLM. https://arxiv.org/abs/2402.15175

[4] Vikrant Varma et al. (2023). “Explaining grokking through circuit efficiency.” https://arxiv.org/abs/2309.02390

[5] Ziming Liu, Eric J. Michaud, & Max Tegmark (2023). “Omnigrok: Grokking Beyond Algorithmic Data.” ICLR. https://arxiv.org/abs/2210.01117

---

> ### Author Response · Authors · 2024-11-16
> **Reply to tfb9's comments**
>
> Thank you for your valuable feedback! We provide response below:
>
> ### **Weakness 1: Lack of Novelty**
> > The findings are somewhat obvious in light of the above citations, and in light of grokking work.
>
> Thank you for pointing out the relevant work. We have added a more clear comparison with existing work on learning syntax. Compared to those works, our work highlights rule learning, training dynamics and random variation. We contrast two types of “grokking” in the existing literature: classic grokking (transition from memorization to generalization) and structural grokking (transition from one generalization rule to another). These transitions occur where the data permits both options to compete; this competition also leads to training instabilities and random seed variation, which are the primary focus of our paper. We provide a more detailed comparison below:
>
> > Mueller et al. (2023) perform a similar analysis, showing that simpler corpora lead to syntactic generalization with less data.
>
> Mueller et al. [1] studies how pretraining data is important for endowing transformers with an inductive bias that favors hierarchical syntactic generalizations. Specifically, they find that pre-training on __simpler__ language, such as child-directed speech, better induces a hierarchical bias than __complex__ language (such as Wikipedia).
>
> In comparison, our first contribution is to identify the __exact syntactic structure__ that induces hierarchical rule learning. Our updated experiments have further narrowed down the relevant source of complexity, pointing to center embeddings specifically, rather than mixed scoping generally. In updated Figure 3 left, we confirm that center embedding sentences induce the hierarchical rule, whereas right branching sentences induce the linear rule.
>
> > At L144, Frank & Mathis (2007) [2] is the correct citation for the syntactic transformations task.
>
>  Thank you for pointing this out; we corrected the citation.
>
> > At L154-161 (and App. F): it would be nice to contextualize this with analyses from McCoy et al. (2020) on instability across random seeds.
>
> McCoy et al. (2020) [2] only mention random seed to briefly point out that the training outcome varies by random seed. In fact, related works nearly always overlook or attempt to control both training instability and seed variability by reporting the average of a few runs. We instead describe the distribution, measuring the probability of selecting a rule or of producing an unstable run.
>
> > In Section 4.3, the findings are related to those of Papadimitriou & Jurafsky (2023) on training with recursive and/or cross-serial dependencies.
>
> Thank you for highlighting this connection—this paper is indeed relevant to hierarchical feature learning, as is Papadimitriou & Jurafsky (2020) [4]. We have added both citations.
>
> ### **Weakness 2: Insights Compared to Grokking**
> > There are probably new insights to be gleaned from this new task setting compared to past grokking work.
>
> Since you are not the only reviewer to be confused about the core contributions and shape of our framework, we have restructured our paper for clarity. Please see our top-level response for the summary of contributions. We use our analysis of data factors in learning to construct a case study of competition between memorization, shortcut generalization rules, and complex generalization rules. It is this precarious competition that leads to both grokking and structural grokking.
>
> To confirm the connection to grokking, in Appendix F we have added additional experiments, showing models trained on low diversity data will memorize syntactic patterns, whereas models trained on high diversity data will extrapolate rules to unseen syntax patterns. This experiment more closely reflects the memorization-to-generalization transition in classic grokking work. However, our paper also addresses the OOD rule competition (Section 5) discussed in more recent *structural* grokking work. We have restructured our paper and expanded the literature review to emphasize these connections.
>
> ### **Weakness 3: Hyperparameters**
>
> > It is not clear why these hyperparameters were chosen, and why different hyperparameters were used for the two tasks.
>
> Similar to classic grokking, the exact condition to induce structural grokking is precarious. The early viability of memorization (for grokking) or linear rules (for structural grokking) is highly sensitive to hyperparameter settings, as has been widely acknowledged in the grokking literature. We copy the setting from existing works to show that even in those specific hyperparameter settings, training can still be highly unstable and final performance can be inconsistent across random seeds.
>
> ### **Weakness 4: Related Work**
> We have updated experiments in Section 4 to focus on the center embedding result, moving other details to the Appendix. Additionally, we expanded the literature review.

---

> ### Author Response · Authors · 2024-11-16
> **Reply to tfb9's questions**
>
> ### **Question 1 (hyperparameters)**
>
> See weakness 3.  Settings are borrowed from prior work.
>
> ### **Question 2 (complexity)**
> We understand that the meaning of "complexity" in our data is somewhat confusing, since we discuss both depth and the presence of center embeddings. The goal of our depth experiments is actually to control for depth and conclusively demonstrate that the relevant factor is instead center embeddings (in the original version we broadly point to mixed scoping, but have updated based on our newer experiments that further narrow it down). We have moved the discussion of depth into the appendix to streamline our results presentation.
>
> ### **Question 3 (studies, clarification)**
>
> It is fairly common to measure the syntactic diversity of a corpus. TED was appropriate for our setting, but some work uses methods like POS-gram counting. As a sample of some papers that measure syntactic diversity using variations on TED, see [5–8], which we will cite as examples. If you know where these practices first originated, we would happily incorporate credit to the original corpus linguistics literature; existing literature reviews are unclear on the history of syntactic diversity measurement.
>
> ### **Question 4 (Fig 4 and Sec 6)**
>
> In figure 1, we differentiate between complexity and diversity. Diversity refers to syntactic diversity (different tree structures). Complexity refers to the presence of center embeddings. While it is true that a mix of right branching and center embedded sentences is also more syntactically diverse, even the 0% center embedding (most simple) example in Fig. 4 has enough diversity to support rule learning over memorization. Therefore, the primary effect of adding more complex examples is to support complex rules over simple rules, as the corpus’s diversity has already made it challenging to memorize all individual syntactic patterns. Another way to think of this is that all models in Fig 4 are already on the right hand side of Fig 1, so even if the added complexity also increases diversity, we are already well outside the memorization regime. The complex rule no longer competes with memorization, but with a systematically applied simple rule.
>
> **References:**
>
> [1] Mueller, Aaron, and Tal Linzen. "How to plant trees in language models: Data and architectural effects on the emergence of syntactic inductive biases." arXiv preprint arXiv:2305.19905 (2023).
>
> [2] McCoy, R. Thomas, Robert Frank, and Tal Linzen. "Does syntax need to grow on trees? sources of hierarchical inductive bias in sequence-to-sequence networks." Transactions of the Association for Computational Linguistics 8 (2020): 125-140.
>
> [3] Papadimitriou, Isabel, and Dan Jurafsky. "Injecting structural hints: Using language models to study inductive biases in language learning." arXiv preprint arXiv:2304.13060 (2023).
>
> [4] Papadimitriou, Isabel, and Dan Jurafsky. "Learning music helps you read: Using transfer to study linguistic structure in language models." arXiv preprint arXiv:2004.14601 (2020).
>
> [5] Ramírez, J., Baez, M., Berro, A., Benatallah, B., Casati, F. (2022). Crowdsourcing Syntactically Diverse Paraphrases with Diversity-Aware Prompts and Workflows. In: Franch, X., Poels, G., Gailly, F., Snoeck, M. (eds) Advanced Information Systems Engineering. CAiSE 2022. Lecture Notes in Computer Science, vol 13295. Springer, Cham. https://doi.org/10.1007/978-3-031-07472-1_15
>
> [6] Song, Yewei, et al. "Revisiting Code Similarity Evaluation with Abstract Syntax Tree Edit Distance." arXiv preprint arXiv:2404.08817 (2024).
>
> [7] Volkart, Lise, and Pierrette Bouillon. "Post-editors as Gatekeepers of Lexical and Syntactic Diversity: Comparative Analysis of Human Translation and Post-editing in Professional Settings." Proceedings of the 25th Annual Conference of the European Association for Machine Translation (Volume 1). 2024.
>
> [8] Gao, N., He, Q. A dependency distance approach to the syntactic complexity variation in the connected speech of Alzheimer’s disease. Humanit Soc Sci Commun 11, 995 (2024). https://doi.org/10.1057/s41599-024-03509-0

---

> > ### Comment · Reviewer_tfb9 · 2024-11-19
> > **Response**
> >
> > Thank you for the detailed response and the very helpful general response. Weaknesses 2 and 4 have been addressed, as have Questions 2 and 4.
> >
> > Weakness 1
> > ===
> > McCoy et al. (2020) link to a project website where the distribution of results across many random seeds are shown for many experiments. See Sections 4 and 5 here: https://rtmccoy.com/rnn_hierarchical_biases.html
> >
> > These results were not part of the original paper, but were linked therein, and I think they refute the idea that this paper was overlooking the impact of random seed. I agree with your more general point that a lot of papers do overlook this or simply report averages, but this bit of nuance would be appreciated, especially as that paper's topic is directly related to this paper's topic.
> >
> > As for Mueller et al. (2023), the distinction between simple/complex language and simple/complex structures is interesting. One would imagine that simpler language should contain simpler syntactic structures on average, whereas more complex language should include more complex structures. Do you believe that the diversity of the complex Wikipedia corpus might interfere with learning robust generalizations? Or is some other factor to blame here? Either way, this could be an interesting discussion point.
> >
> > Weakness 3
> > ===
> > Could you cite the specific "existing works" whose hyperparameters you use? This still feels pretty vague.
> >
> > Question 1
> > ===
> > Same as Weakness 3.
> >
> > Question 3
> > ===
> > Thank you. These citations will be sufficient for establishing that this approach has been used in prior work, albeit for different purposes. I also see that the language in the paper has already changed to no longer refer to prior linguistics studies when you define the syntactic diversity metric; this is also a welcome change which makes it less pressing to locate these prior studies. If you would like a further citation that uses tree edit distance specifically to measure syntactic diversity, consider citing [1]. You could also contrast this with past approaches, like the POS n-gram approach you mention (used in [2], for example). I also cannot find the first paper to do either of these, as neither cite the inspiration for this technique. I think some combination of these citations should probably be sufficient.
> >
> > Once these and the experiments referred to in the new Sec. 6.1 are added, I will consider this fully addressed.
> >
> > Review changes
> > ===
> > I am changing my score to a 5 to reflect the addressed points, and in anticipation of the new experiments and citations. While I still have concerns regarding vague sources for important methodological details and incremental findings (extending prior findings to a new task setting, and attributing known behaviors to particular syntactic structures, which would not be likely to be controllable in a real pre-training setting), I think that the contributions of this work are now much clearer, and much better contextualized with past work. The sheer quantity of results on this topic are also impressive and potentially worthwhile to the set of researchers studying syntactic generalization and syntactic grokking, even if the implications for interpretability/evaluation/grokking researchers and LM engineers more broadly are currently unclear.
> >
> > Also, I apologize for not including the references in my original review. I have edited my review to add them. It would be nice to add [3-4] in particular when contextualizing this work with grokking findings; I see you already have [5] in your general response.
> >
> > References
> > ===
> > [1] Kuan-Hao Huang et al. (2023). "ParaAMR: A Large-Scale Syntactically Diverse Paraphrase Dataset by AMR Back-Translation." ACL.
> >
> > [2] Shen-yun Miao et al. (2020). "A Diverse Corpus for Evaluating and Developing English Math Word Problem Solvers." ACL.

---

> ### Author Response · Authors · 2024-11-20
> **Reply to tfb9 1/2**
>
> We want thank reviewer tfb9 for the timely and thoughtful comments! We address specific responses below:
>
> ### **Weakness 1**
> > I agree with your more general point that a lot of papers do overlook this or simply report averages, but this bit of nuance would be appreciated, especially as that paper's topic is directly related to this paper's topic.
>
>
> Thank you for highlighting this supplementary report! We have added a citation to this blog post in our related work section, as we agree that including this reference adds valuable nuance to our main contributions.
> In the post, they specifically noted: “Note that there is considerable generalization variability across reruns of the same model; e.g., the SRN with content-based attention had generalization set first-word accuracy ranging from 0.17 to 0.90.” While this acknowledges the significance of variability across random seeds, their approach primarily relies on summary statistics, stating: “To understand models' typical behavior, therefore, we mainly consider the median across all 100 initializations.”
>
> In contrast, our work explicitly examines the distribution of outcomes and identifies key properties, such as the bimodal or clustered nature of solutions (e.g., Figure 4’s horseshoe plots). This focus on distributional properties differentiates our approach. That said, we have cited this prior work to acknowledge their visualization of variances across runs, which aligns with and complements our findings.
>
> > As for Mueller et al. (2023), the distinction between simple/complex language and simple/complex structures is interesting. One would imagine that simpler language should contain simpler syntactic structures on average, whereas more complex language should include more complex structures. Do you believe that the diversity of the complex Wikipedia corpus might interfere with learning robust generalizations? Or is some other factor to blame here? Either way, this could be an interesting discussion point.
>
> Thank you for raising this interesting question. Since our notion of complexity is based specifically on structural features requiring hierarchical modeling (e.g., embedded clauses), we cannot definitively say whether the Wikipedia corpus contains more embedded clauses than child-directed speech (CHILDES). However, it is reasonable to assume that Wikipedia includes a greater diversity of syntactic structures.
> We think it is likely that the Wikipedia is “complex” in two ways:  task is more complex and language is more complex and both factors make Wikepdia data harder to learn grammar from :
>   * Task complexity: Training on Wikipedia likely requires models to learn not only syntax but also world knowledge and other properties, whereas simpler corpora like CHILDES primarily emphasize linguistic structure. As Mueller et al. (2023) noted, smaller language models may struggle to develop robust language capabilities due to the added complexity of multitask learning.
>   * Lexical complexity: The vocabulary size of Wikipedia (\~10M) is significantly larger than that of CHILDES (\~10K). A smaller vocabulary simplifies the distribution of part-of-speech (POS) patterns, likely making CHILDES dataset easier for models to acquire grammatical capabilities.
>
> While Mueller et al. (2023) discuss the second factor in their analysis, their setting—though less synthetic—does not allow for granular, controlled experiments to isolate these explanations. In our framework, we focus on a simpler language modeling task using data generated by context-free grammars with a fixed vocabulary size. This setup allows us to define specific metrics and measure distinct properties of the training data, enabling a deeper understanding of model behavior in a highly controlled environment.
>
> ### **Weakness 3**
> > Could you cite the specific "existing works" whose hyperparameters you use? This still feels pretty vague.
>
> Absolutey, here is a list of existing works that uses the same hyperparameters. We have updated citation in the PDF as well! Thank you for pointing this out!
>
> [1] Murty, Shikhar, et al. "Grokking of hierarchical structure in vanilla transformers." arXiv preprint arXiv:2305.18741 (2023).
>
> [2] Ahuja, Kabir, et al. "Learning Syntax Without Planting Trees: Understanding When and Why Transformers Generalize Hierarchically." arXiv preprint arXiv:2404.16367 (2024).
>
> ### **Question 3**:
> Thank you for the additional insights, we have revised Section 6.1 adding the additional citations you have provided!
>
> >  Once these and the experiments referred to in the new Sec. 6.1 are added, I will consider this fully addressed.
> Are you referring to the experiments showing that when the model is exposed to a fewer unique syntax trees during training, it memorizes those patterns without extrapolating any rules to unseen sentence structures? You can find experiment in Appendix F. If not, could you help us by pointing out which experiments you are referring to? Thank you!

---

> ### Author Response · Authors · 2024-11-20
> **Reply to tfb9 2/2**
>
> ### **Review changes**
> > The sheer quantity of results on this topic are also impressive and potentially worthwhile to the set of researchers studying syntactic generalization and syntactic grokking, even if the implications for interpretability/evaluation/grokking researchers and LM engineers more broadly are currently unclear.
>
> Thank you for recognizing the relevance of our results to the literature on syntactic generalization and syntactic grokking. We hope that our findings will be valuable to researchers focusing on the linguistic capabilities of language models.
>
> Regarding your concern about unclear implications for interpretability, evaluation, and grokking researchers, we believe that the second half of our work has broader implications for interpretablity/grokking community. Specifically, in more general settings, works such as [1] and [2] have shown that during training, different circuits or heuristics compete, and models trained on different random seeds can lead to distinct generalization behaviors depending on which circuits dominate. While these works primarily attribute generalization differences to circuit formation, our findings highlight that this competition between circuits can also destabilize learning dynamics.
>
> Moreover, we show that these unstable runs obscure the true distribution of model behaviors, making it appear less bimodal than it is. Once unstable OOD behaviors are accounted for, we find that clustered capabilities—representing stable generalization strategies—might be more prevalent than previously recognized. This insight into clustering dynamics could offer practical value to LM engineers aiming to better understand and predict model behaviors.
>
> For the broader grokking community, our results extend the analysis of competition between memorization and generalization (as studied in [2]) to the competition between distinct generalization rules. This suggests that a deeper understanding of competing circuits is not only critical for understanding dynamics grokking but could also illuminate breakthrough capabilities in language models, such as those observed in tasks like MMLU.
>
>
> > It would be nice to add [3-4] in particular when contextualizing this work with grokking findings
> We have updated the related work on grokking to include these citations along with the ones mentioned above.
>
>
> [1] Bhaskar, Adithya, Dan Friedman, and Danqi Chen. "The Heuristic Core: Understanding Subnetwork Generalization in Pretrained Language Models." arXiv preprint arXiv:2403.03942 (2024).
>
> [2] Huang, Yufei, et al. "Unified view of grokking, double descent and emergent abilities: A perspective from circuits competition." arXiv preprint arXiv:2402.15175 (2024).

---

### Author Response · Authors · 2024-11-16
**Meta-Response**

We want to thank all the reviewers for their time and valuable feedback! Because some reviewers were unclear on what our novel contributions are, we have realized that our presentation does not sufficiently clarify our new framework or results. We have revised the paper to explain this framework more clearly, and map it out below.

Previous work has investigated the role of diversity in promoting generalization over memorization. Other work has also investigated some data factors that influence the selection of one generalization rule over another. However, none of these works unify all our three themes: (1) complexity AND diversity, (2) memorization/generalization AND rule selection, (3) training stability AND consistency across seeds. This relationship can be summarized through a series of points:

* Diverse data promotes general rules over memorization. This is an extension of existing results in grokking and compositionality to a new setting (i.e., structural grokking), with new precision over the definition of diversity (in our case, latent structure diversity).
  * In our updated version, we have supported and refined this point by confirming that in the memorization regime, models still apply rules consistently to familiar syntactic structures. In the generalization regime, by contrast, models extrapolate those rules to unseen syntactic structures.

* Complex data promotes complex rules (i.e., rules that rely on latent structure). This is an extension of existing results in OOD generalization from ambiguous training data, though again we have controlled the setting more precisely and found more specific complexity factors (center embeddings) driving hierarchical generalization.

  * The revised paper further narrows down the relevant complexity factor, from mixed scoping broadly to center embeddings specifically (i.e., the "degree 2" embedding structures that were previously only addressed in the discussion section). We have added experiments confirming that center embeddings explain rule selection better than other sources of syntactic complexity.

* Intermediate Diversity and Complexity: Intermediate diversity creates an unstable regime where memorization and rule-based strategies compete. This intermediate diversity regime is a novel discovery, and may be considered surprising, since it is not obvious that intermediate diversity will lead to more instability than either heterogeneous or homogeneous data. Similarly, intermediate complexity promotes instability due to competition between generalization rules. These findings support prior work on oscillatory dynamics and subset competition (e.g., Elan et al. [1]).

Furthermore, we have connected these unstable training dynamics and data properties to the much-overlooked role of random variation across seeds. While some existing work discusses random variation in generalization rules [2] or in the timing of grokking [3], to our knowledge, this is the first paper to make a direct connection between clustered OOD behavior and the likelihood of unstable training. This connection yields the horseshoe-shaped curves in Fig. 4. In particular, we show:

* Stable training runs are correlated, across random seeds, with either memorization or systematic application of one of the generalization rules. Models that do not commit to one of these strategies---memorization, linear rule, or hierarchical rule---tend to be unstable.

  * This might be an extension of existing findings: omnigrok [4] shows that in their specific setting, generalization is associated with more stability, but they don't show that memorization can also be stable. They also do not consider correlation across random seeds, meaning that this stability could be a direct effect of their regularizer rather than representing rule commitment more generally under natural variation.

* Data settings that create a higher likelihood of unstable training runs also allow models to learn either rule, determined only by the random initialization seed.

  * This finding addresses a missing gap in the literature. Given findings like Juneja et al. [2]  and Zhou et al. [5], it has thus far been unclear why some settings lead to inconsistent OOD behavior across random seeds. Our findings confirm that the inconsistency is a direct result of intermediate levels of data complexity or data diversity that promote different model strategies.

  * Our work also shines light on why clusters of OOD behavior can form across random seeds, per juneja et al [2]. Our work illustrates that these clusters have formed around stable model strategies and that training runs are unstable unless they find these basins.

  * Clustering effects tie to a number of key ideas in recent literature. Sharp transitions between generalization rules lead to phenomena like emergence, where sudden breakthroughs occur because there is no way to interpolate between rules.

We have emphasized the connections above in our updated presentation.

---

> ### Author Response · Authors · 2024-11-16
> **References**
>
> [1] Rosenfeld, Elan, and Andrej Risteski. "Outliers with opposing signals have an outsized effect on neural network optimization." arXiv preprint arXiv:2311.04163 (2023).
>
> [2] Juneja, Jeevesh, et al. "Linear connectivity reveals generalization strategies." arXiv preprint arXiv:2205.12411 (2022).
>
> [3] Hu, Michael Y., et al. "Latent state models of training dynamics." arXiv preprint arXiv:2308.09543 (2023).
>
> [4] Liu, Ziming, Eric J. Michaud, and Max Tegmark. "Omnigrok: Grokking beyond algorithmic data." The Eleventh International Conference on Learning Representations. 2022.
>
> [5] Zhou, Yongchao, et al. "Transformers can achieve length generalization but not robustly." arXiv preprint arXiv:2402.09371 (2024).

---

### Meta-Review · Area_Chair_1Myo · 2024-12-20

**Metareview:**

This paper studies a particular aspect of how the composition of the training data influences the generalization (vs memorization) behavior of language models. Reviewers appreciated the study of inductive biases of language models. However, there was generally a lack of clarity on exactly what the contributions here are. In their response, the authors have attempted to clarify this. However, this seems to represent a major revision, requiring a resubmission of the paper.

**Additional Comments On Reviewer Discussion:**

The reviewers were unclear on the claimed contribution. The authors acknowledge this and propose a new framing. However, the rework seems to rise to the level of a major revision, and thus the paper should be rewritten and submitted to another conference.

---

### Decision · Program_Chairs · 2025-01-22

Reject